# Effects of Copper on *Legionella pneumophila* Revealed via Viability Assays and Proteomics

**DOI:** 10.3390/pathogens13070563

**Published:** 2024-07-03

**Authors:** Yang Song, Didier Mena-Aguilar, Connor L. Brown, William J. Rhoads, Richard F. Helm, Amy Pruden, Marc A. Edwards

**Affiliations:** 1Civil and Environmental Engineering, Virginia Tech, 1145 Perry St., 418 Durham Hall, Blacksburg, VA 24061, USAedwardsm@vt.edu (M.A.E.); 2Utilities Department, 316 N. Academy St., Town of Cary, Cary, NC 27513, USA; 3Biochemistry, Virginia Tech, 340 W Campus Dr, Blacksburg, VA 24060, USA; 4Department of Biochemistry, University of Nebraska-Lincoln, N106, The Beadle Center, Lincoln, NE 68588, USA; 5Genetics, Bioinformatics, and Computational Biology, Virginia Tech, Steger Hall, Blacksburg, VA 24061, USA; 6Black & Veatch, 8400 Ward Pkwy, Kansas City, MO 64114, USA; 7Department of Biochemistry, Virginia Tech, 1015 Life Science Circle, 211B Steger Hall, Blacksburg, VA 24061, USA; helmrf@vt.edu

**Keywords:** *Legionella pneumophila*, copper, viability, proteomics, amoebae

## Abstract

Cu is an antimicrobial that is commonly applied to premise (i.e., building) plumbing systems for *Legionella* control, but the precise mechanisms of inactivation are not well defined. Here, we applied a suite of viability assays and mass spectrometry-based proteomics to assess the mechanistic effects of Cu on *L. pneumophila*. Although a five- to six-log reduction in culturability was observed with 5 mg/L Cu^2+^ exposure, cell membrane integrity only indicated a <50% reduction. Whole-cell proteomic analysis revealed that AhpD, a protein related to oxidative stress, was elevated in Cu-exposed *Legionella* relative to culturable cells. Other proteins related to cell membrane synthesis and motility were also higher for the Cu-exposed cells relative to controls without Cu. While the proteins related to primary metabolism decreased for the Cu-exposed cells, no significant differences in the abundance of proteins related to virulence or infectivity were found, which was consistent with the ability of VBNC cells to cause infections. Whereas the cell-membrane integrity assay provided an upper-bound measurement of viability, an amoebae co-culture assay provided a lower-bound limit. The findings have important implications for assessing *Legionella* risk following its exposure to copper in engineered water systems.

## 1. Introduction

The efficacy of *Legionella* control measures is typically assessed using culture enumeration methods, such as culturing with buffered charcoal yeast extract (BCYE) agar [1]. However, some *Legionella* that do not grow on traditional culture media have been shown to still be viable and capable of infecting protozoa [2,3,4,5,6,7], infecting human macrophages [3], or regrowing in the water system when conditions are again suitable [8]. Cells that are still technically viable but unable to be cultured on standard media are referred to as viable but non-culturable (VBNC). Given that VBNC *Legionella* could still potentially pose a human health risk [9], it is important to build an understanding of the conditions that induce a VBNC state in engineered water systems. Further, there is a need to develop and refine techniques for detecting and enumerating VBNC *Legionella*.

Much of the research focused on VBNC *Legionella* to date has been based solely on measurements of membrane integrity [10,11,12,13,14,15]. Other viability tests that have been implemented include measurement of enzyme activity, electron transfer activity, mRNA production, and ATP production [3,4,16,17]. Some viability test parameters, such as enzyme activity and total ATP levels, were reported to be better indicators for cell viability/activity [14]. However, all of these alternative viability tests are still susceptible to overestimation of the proportion of *Legionella* cells that maintain their infectivity [3,4,16]. Alternatively, “resuscitation” assays have been applied, where samples containing non-culturable *Legionella* are mixed with amoeba cultures to determine if the presence of a host organism can render *Legionella* in the sample culturable [18]. Such an approach is conservative, providing a lower-bound estimate of the number of viable cells. However, amoebae resuscitation assays are sensitive to the co-culture conditions, including the multiplicity of infection (MOI) used (i.e., the ratio of amoeba to *Legionella* combined), the co-culture time (range = 2–22 days), and co-culture temperature (range = 25–37 °C) [2,3,4,5,6,16].

Characterizing the metabolic response of *Legionella* to external stress, such as disinfection, could provide powerful insight into whether it induces a VBNC state in *Legionella*, by what mechanisms, and whether virulence is maintained. Disinfection is often applied to water systems, in particular, plumbing conveying drinking water and hot water throughout buildings (i.e., premise plumbing), for the purpose of controlling *Legionella* and other pathogens. Commonly applied disinfectants and other environmental controls, including chlorine, chloramine, ultraviolet irradiation, temperature shock, and starvation, have been shown to induce VBNC *Legionella* [2,3,4,6,7,19,20] (Figure 1). There is also increasing interest in the application of copper (Cu) as a biocide, which can be dosed purposefully, often as part of a copper–silver ionization system, or employed passively through the natural release of copper from pipes as a result of corrosion. A recent study also indicated that *L. pneumophila* Cu tolerance might vary in realistic plumbing systems [21].

Examining the whole-cell protein abundance (i.e., the proteome) of Cu-exposed *Legionella* cells relative to other measures of virulence and infectivity could also provide insight into processes associated with the putative VBNC state. However, few attempts have been made to characterize the proteome of VBNC *Legionella*. One study demonstrated that monochloramine-induced VBNC *Legionella* could still produce virulence-associated proteins, but their levels were insufficient to retain the infectivity of the non-culturable cells, as indicated by the inability to resuscitate them with amoebae co-culture [16]. Another study demonstrated that virulence-related proteins could be recovered from non-culturable *Legionella* cells, but co-culturing with amoebae was not attempted [22].

To our knowledge, neither the potential for Cu to induce VBNC *Legionella* nor differences in metabolic response to Cu stress has been examined among *Legionella* strains. If Cu is, in fact, found to induce VBNC *Legionella*, this may also help to explain conflicting observations that sometimes Cu pipe, compared with plastic pipe materials, is associated with a higher level of *Legionella* detected by qPCR but not culture [23].

The objective of this study was to apply a suite of viability assays and proteomics to assess the effects of Cu on *L. pneumophila* and to evaluate evidence for the production of a VBNC state and the ability to infect cells. Specifically, we tested two strains of *L. pneumophila*, one with high and one with low tolerance of Cu, comparing both Cu-exposed and Cu-free control cells. The findings can help to better understand the limits of copper as an antimicrobial for the control of *Legionella* in water systems and to inform improvement in methods for monitoring the efficacy of disinfectants in premise plumbing.

## 2. Methods

### 2.1. Legionella pneumophila Strain, Sample Preparation, and Culture

*L. pneumophila* cultures were freshly prepared for each experimental run using serogroup 1 strain 130 b obtained from the Centers for Disease Control and serogroup 1 environmentally isolated outbreak-associated strain from Quincy, IL, stored at −80 °C in buffered yeast extract (BYE) broth (10 g ACES, 10 g yeast extract, 1 g α-ketoglutaric acid, 0.4 g L-cysteine and 0.25 g ferric pyrophosphate per litter; Sigma-Aldrich, St. Louis, MO, USA) with glycerol (20% *v*/*v*). Frozen stocks were isolated onto buffered charcoal–yeast extract (BCYE) agar plates (12 g agar, 2 g activated charcoal, 1 g α-ketoglutarate monopotassium, 10 g ACES buffer, 2.8 g potassium hydroxide, 0.25 g iron pyrophosphate, 3 g ammonium-free glycine, 80,000 IU polymyxin B sulfate, 0.001 g vancomycin hydrochloride, 0.08 g cycloheximide, 0.4 g L-cysteine monohydrate per litter; Sigma-Aldrich, St. Louis, MO, USA) and incubated for 72 h at 37 °C. BYE broth was inoculated with *L. pneumophila* to a target initial optical density at wavelength 600 nm (OD_600_) of 0.2 by transferring 1 mL of sterile BYE broth onto t-streaked culture plates and detaching colonies with a sterilized glass cell spreader. Inoculated BYE broth was incubated at 37 °C with agitation for 14 h to reach the early stationary phase, as determined by OD_600_ growth curves [24,25]. After broth culture, cells were pelleted by centrifuging at 5000× *g* for 10 min. The base water used in this study was prepared by passing Blacksburg tap water through GAC to remove disinfectant residuals, ferric oxide (Brightwell Aquatics, Fort Pane, AL, USA) filters to remove orthophosphate, followed by filter sterilization and autoclaving. Cell pellets were washed twice with 10 mL base water, resuspended in 2 mL base water, and diluted to approximately 3 × 10^8^ CFU/mL as a stock solution (McFarland Standard No. 1, OD600 = 0.257).

The International Organization of Standardization Draft International Standard 11731 was followed to culture *Legionella.* Briefly, samples were plated (using serial dilution technique [25] or 0.1–1 mL, depending on dilution required) onto BCYE media. Putative *Legionella* isolates were streaked onto BCYE media with and without 0.4 g/L of L-cysteine for culture confirmation. All samples were plated in triplicate for enumeration of CFU.

### 2.2. Cu Inactivation Experiments

Copper stock solutions (100 mg/L copper as CuSO_4_) were freshly prepared and adjusted to pH = 4.0 to maintain copper solubility. Cu was quantified using a Thermo Electron X-series inductively coupled plasma spectrometer (Thermo Fisher Scientific, Waltham, WA, USA) per Standard Method 3125-B [26] and EPA-approved colorimetric method [27]. Cu inactivation test procedures were similar to our previous protocol [25]. Briefly, after warming to 37 °C, base water pH was adjusted to 6.5, followed by a Cu dose of 0 or 5 mg/L, pH adjustment back to 6.5, if necessary, and mixing for 20 min to allow copper speciation to stabilize. Glass bottles were inoculated in a biosafety cabinet with *L. pneumophila* at ~3 × 10^7^ and ~3 × 10^6^ CFU/mL for 130 b and the outbreak-associated strain, separately. After mixing, initial culturable and viable *L. pneumophila* levels were quantified as described in the culture method, and solutions were transferred to sterile 50-mL sacrificial glass reactors and incubated at 37 °C. Reactors were inverted to mix and sacrificed to quantify culturable, viable levels of *L. pneumophila* at t = 4, 24 h for 130 b and outbreak strain (sampling intervals and initial cell densities were determined based on previous test runs to achieve <1 CFU/mL culturability). Samples containing Cu were mixed with EDTA (Thermo Fisher Scientific, Waltham, WA, USA) at a 5:1 (EDTA:Cu) molar ratio to quench the antimicrobial effects of Cu before culture and viability tests. For co-culture with amoebae, besides mixing with EDTA solution to prepare the *Legionella* cell suspension, another EDTA washing protocol was proven to remove >90% Cu from cell pellets was implemented for comparison (Appendix A). The EDTA washing procedures included centrifugation of the cell suspension at 5000× *g* for 10 min, decanting the supernatant, washing the pellet with 10 mL EDTA solution at 5:1 molar ratio to complex Cu, and then sitting for 30 min. The suspension was centrifuged again at 5000× *g* for 10 min, and the pellet was resuspended with 5 mL base water to prepare a stock solution for co-culture tests. Biological replicates were obtained by repeating the entire process three or four times.

### 2.3. Viability Assays

*Legionella* viability assays included two fluorescence-based and one luminescence-based viability protocols. The fluorescence-based protocols included (1) membrane integrity—Live/Dead BacLight viability kit (Thermal Fisher Scientific, Waltham, MA, USA) double staining with 3 µL 20 mM propidium iodide and 3 µL of 3.34 mM Syto-9 being added to 2 mL sample, followed by 15 min incubation at room temperature based on manufacturer’s instruction and (2) esterase activity—6-carboxyfluorescein diacetate (CFDA) (Sigma-Aldrich, St. Louis, MO, USA) staining with 2.5 µL of 10 mM stock 6-CFDA being added to 500 µL samples with 50 µL of 10 mM EDTA, followed by a 30 min incubation at 35 °C [4,28].

The luminescent BacTiter-Glo^TM^ Microbial Cell Viability Assay kit (Promega, Madison, WI, USA) was used to quantity the intracellular ATP concentrations following manufacturer’s instruction after 100 µL of luminescence reagent was added to 100 µL cell suspensions in an opaque 96 well plate. The relative ATP concentration was determined by comparing the cellular luminescence values to a standard curve with known ATP concentrations. The highest luminescence of each well was recorded by using a multi-mode microplate reader (Spectramax M5, Molecular Devices, San Jose, CA, USA), and the results were analyzed in Softmax Pro (v 7.1).

### 2.4. Flow Cytometry

FCM analyses were performed with a BD Acuri C6 flow cytometer (BD Bioscience, Franklin Lakes, NJ, USA) and data analysis software CSampler (Version 1.0.264.21). Forward scatter (FSC), side scatter (SSC), and four fluorescence signals were measured. A 518–548 nm band pass filter was used to collect the green fluorescence (FL1); a 565–605 nm band pass filter was used to collect the yellow–orange fluorescence (FL2); a >670-nm band pass filter was used to collect the red fluorescence (FL3), and a 655–700 nm band pass filter was used to collect the red fluorescence (FL4). The laser excitation wavelengths were 488 and 640 nm for FL1-3 and FL4, separately. In the experiment performed with *L. pneumophila*, the FCM analyses were performed with detector settings, including a medium flow speed (50 µL/min), a threshold FSC signal of 800, and FL1 signal of 800. Based on the fluorescence excitation and emission wavelengths, the cell membrane integrity plot was analyzed with the FL1 vs. FL3 plot, and enzyme activity was analyzed with the FSC vs. FL1 plot.

### 2.5. Proteomic Sample Preparation and Mass Spectrometric Analyses

Triplicates of strain 130 b control and Cu = 5 mg/L cell suspension at ~3 × 10^7^ CFU/mL were prepared for proteomic analyses after four hours of incubation at 37 °C. Cell suspensions were centrifuged at 5000× *g* for 15 min, resuspended in 9:1 methanol:water, and incubated overnight at −20 °C, followed by centrifugation at 12,000× *g* for 15 min. Afterward, pellets were resuspended in lysis buffer (pH = 7.55) containing 5% sodium dodecylsulfate (Fisher Scientific, Waltham, MA, USA) and 50 nM triethylammonium bicarbonate (TEAB) (MilliporeSigma, St. Louis, MO, USA). Dithiothreitol (MilliporeSigma, St. Louis, MO, USA) was added to a final concentration of 20 mM, followed by 10 min incubation at 95 °C and addition of iodoacetamide (MilliporeSigma, St. Louis, MO, USA) to a final concentration of 40 mM after cooling to room temperature, followed by 30 min incubation in the dark at room temperature and then addition of o-phosphoric acid (MilliporeSigma, St. Louis, MO, USA) to a final concentration of 1.2%. Samples were further processed using S-Trap™ micro spin columns following the suggested protocol [29]. In brief, six volumes of S-trap binding buffer (90% MeOH, 100 mM final TEAB, pH 7.1) were added to the lysate, and the mixture was passed through the micro spin column by centrifugation (4000× *g*, 1 min). The column was washed three times with 150-μL of S-Trap binding buffer. Proteins were then digested to peptides by adding trypsin (Fisher Scientific, Waltham, MA, USA) at a 1:25 ratio (trypsin:protein, weight basis) directly to the micro spin column and incubated at 37 °C overnight. Peptides were eluted in four steps: first with 40 μL of 50 mM TEAB, followed with 40 μL of 0.2% (*v*/*v*) aqueous formic acid (MilliporeSigma, St. Louis, MO, USA), and finally twice with 35 μL of 50% acetonitrile containing 0.2% formic acid. Eluted fractions were pooled and dried down in a Centrivap concentrator (Labconco Corporation, Kansas City, MO, USA) at 45 °C. Samples were then resuspended in acetonitrile:water (2:98, *v*/*v*) that contained 0.1% formic acid (*v*/*v*) to a final concentration of 1 μg/μL and filtered through a 0.2 um PES filter vial (Thomson Instrument Companies, Clear Brook, VA, USA). For LC/MS analyses, 18 μg of digested protein was injected per sample.

Samples were first loaded onto a precolumn [Acclaim PepMap 100 (Thermo Scientific, Waltham, MA, USA), 100 µm × 2 cm], after which flow was diverted to an analytical column [50 cm µPAC (PharmaFluidics, Woburn, MA, USA)]. The UPLC/autosampler utilized was an Easy-nLC 1200 (Thermo Scientific, Waltham, MA, USA). The flow rate was maintained at 150 nL/min, and peptides were eluted utilizing a 2–10% gradient of solvent B in solvent A over 3 min, followed by a 10–50% gradient of solvent B in solvent A over 88 min. The mass spectrometer utilized was an Orbitrap Fusion Lumos Tribid^TM^ (Thermo Scientific, Waltham, MA, USA). The spray voltage on the µPAC compatible Easy-Spray emitter (PharmaFluidics, Woburn, MA, USA) was 1300 volts; the ion transfer tube was maintained at 275 °C; the RF lens was set to 30%, and the default charge state was set to 2. MS data for the *m*/*z* range of 400–1500 were collected using the Orbitrap at 120,000 resolution in positive profile mode with an AGC target of 4 × 10^−5^ and a maximum injection time of 50 ms. Peaks were filtered for MS/MS analysis based on having isotopic peak distribution expected of a peptide with an intensity above 2 × 10^−4^ and a charge state of 2–5. Peaks were excluded dynamically for 15 s after 1 scan with the MS/MS set to be collected at 45% of a chromatographic peak width with an expected peak width (FWHM) of 15 s MS/MS data starting at *m*/*z* of 150 were collected using the orbitrap at 15,000 resolution in positive centroid mode with an AGC target of 1.0 × 10^5^ and a maximum injection time of 200 ms. The activation type was HCD, which stepped from 25 to 35.

Data analysis utilized Proteome Discoverer (PD, Version 2.5; Thermo Scientific Waltham, MA, USA), combining a Sequest HT and Mascot 2.7 (Matrix Science, Boston, MA, USA) search into one result summary for each sample. Databases used in the analysis were the Uniprot reference proteome *L. pneumophila* and a common protein contaminant database provided with the Proteome Discoverer (PD) software package. Each search assumed trypsin-specific peptides with the possibility of one missed cleavage, a precursor mass tolerance of 10 ppm, and a fragment mass tolerance of 0.8 Da. Sequest HT searches included a fixed modification of carbamidomethyl at Cys and the following dynamic modifications: oxidation of Met; acetylation of the protein *N*-terminus; Met loss of the protein *N*-terminus; and Met loss with acetylation of the protein *N*-terminus. Protein identifications were reported at a 1% false discovery rate (high confidence) or at a 5% false discovery rate (medium confidence) based on searches of decoy databases utilizing the same parameters as above. The software matched peptide peaks across all runs and protein quantities, which are the sum of all peptide intensities associated with the protein. Normalization was based on the assumption that total peptide amounts injected for each analysis were equal. Specific proteins of virulence factors, stress response, and heavy metal transport were identified by mapping the manually curated gene ontology (GO) databases based on categories of interest. For the category of “Cu related”, “oxidoreductive process”, and “amoebae interaction/Infectivity”, key words “copper”, “redox/oxido/reductase/superoxide”, “amoeba/secretion system/infect” were used separately to screen out GO IDs of interest from the full GO ID list accessed from geneontology.org. All the GO IDs were also manually confirmed to be involved in the prokaryotes. Protein abundances within the categories of interest were compared between Cu-exposed non-culturable and Cu-free culturable *L. pneumophila* conditions. For the cluster of orthologous groups (COG) categories identification analysis, specific protein abundances were compared at the beginning and end of the experiments for both copper exposure and control conditions. The proteins with significant abundance changes were identified by LefSe analysis [30] using a significance level of 0.05 and LDA score of 2.0, further analyzed through EggNOG [31] to identify the Cluster of orthologous groups (COG) categories.

### 2.6. Amoebae Co-Culture Experiments

Immediately after *L. pneumophila* cell suspensions were prepared for culturability and viability tests, split samples were processed for co-culture with *Acanthamoeba castellanii*.

*A. castellanii* (ATCC-30234) obtained from the Centers for Disease Control was maintained in 25 cm^3^ tissue culture flasks containing 10 mL axenic proteose yeast glucose (PYG) broth culture medium (ATCC medium 712) at 25 °C. *A. castellanii* was maintained by passaging cells with new sterile PYG medium bi-weekly.

Replicate culture tubes containing *A. castellanii* were prepared by inoculating each 15 mL culture tube with an exponential phase (3–4 days growth in fresh PYG) amoeba culture in 2-mL PYG medium to reach a final concentration of ~1 × 10^4^ cells/mL. *L. pneumophila* were harvested and added to each replicate tube corresponding to a final multiplicity of infection (MOI) of 100, 10, 0.01, 0.001 and 10, 0.01, 0.001 for *L. pneumophila* 130 b and outbreak-associated strain, respectively, per *A. castellanii*, followed by slight centrifugation at 1000× *g* for 1 min and vortex before incubation at 25 °C. Positive control (healthy *L. pneumophila*) and negative control (no dose of *L. pneumophila*) were included in each co-culture experiment. On days 7 and 14, co-culture cells were harvested by scraping the monolayers using a sterile cell scraper (Celltreat Scientific, Pepperell, MA, USA), and *L. pneumophila* was separated from amoebae by physical passage through a sterile gauge 20 needles 10–15 times. Due to the lower starting cell density for outbreak-associated strain, the highest MOI ratio was only set as 10.

Two approaches were applied to mitigate potential interference of Cu with the assay in Cu-exposed conditions: (1) directly mixing with EDTA at 1:5 (Cu: EDTA) molar ratio (Cu-EDTA) prior to co-culture with amoeba at a MOI of 1:100 (*Legionella*: amoeba) for strain 130 b and 1:10 for the outbreak-associated strain; (2) mixing samples with EDTA solution, allowing for 30 min reaction time, and centrifuging the sample to pellet the cells to remove >90% copper (Appendix A) prior to co-culture at the same MOIs (Cu-EDTA Washing).

### 2.7. Data Analysis

Summary statistics (mean, standard deviation) and graphs were generated using the R (version 3.6.1) and Microsoft Excel 2016 (Microsoft). Venn diagrams were created using the R package “VennDiagram”. Statistical comparisons of the protein abundances among groups of samples were conducted in LEftSe [30]. Functional annotations of the proteins were accomplished through eggnog-mapper (http://eggnog-mapper.embl.de (accessed on April 2022)) [31]. GO IDs for the proteins of interest were retrieved using the R package “Uniprot” with the function of “GetProteinGOInfo”. All raw proteomics data are available at ProteomXchange (accession Number PXD51925).

## 3. Results

### 3.1. Copper Caused Loss of Culturability at High Doses

*L. pneumophila* culture levels remained stable (<one-log change) in the control (i.e., copper-free) sterile base water at pH = 6.5 for both strains. There were also no significant changes in parameters that served as indicators for viability for either strain, with only a 2–22% decrease in cell membrane integrity and a 3–15% decrease in enzyme activity (Figure 2). After 4 and 24 h exposure to 5 mg/L Cu, the culturability of strain 130 b and the outbreak-associated strain was reduced from a starting cell density of ~3 × 10^7^ CFU/mL and ~3 × 10^6^ CFU/mL, respectively, to less than 10 CFU/mL (Appendix A). The lower cell initial cell density of 3 × 10^6^ CFU/mL was applied to the outbreak-associated strain because it was more copper resistant [32] and, as a result, >300 CFU/mL culturability remained after 48 h of incubation with 5 mg/L Cu when the initial cell density was 3 × 10^7^ CFU/mL.

### 3.2. Non-Culturable Cells Retained Indicators of Viability

Despite the five- to six-log reduction in culturable *Legionella*, both strains retained other measures of viability when dosed with 5 mg/L Cu. Cell membrane integrity remained 50–64% intact (Appendix A), and enzyme activity decreased by <40% (Appendix A). While ATP decreased in strain 130 b by 94%, it only decreased in the outbreak-associated strain by 46%. It is worthwhile to note that the ATP levels generated by the copper-exposed outbreak-associated strain at the end of incubation were comparable to the dead cell controls (2.93 × 10^−15^ vs. 2 × 10^−14^ µM ATP; Appendix A).

### 3.3. Proteomics of Cu-Exposed L. pneumophila

Over 1600 proteins were detected across both the 130 b Control and Cu-exposed conditions. A total of 97.6% of the proteins were shared between the control and copper-exposed non-culturable *L. pneumophila* (Figure 3, Appendix A). The 32 unique proteins identified in the Cu-exposed *Legionella* condition were functionally categorized as “transport”, “glycerol metabolic process”, “response to heat”, “transmembrane transporter activity”, “binding activity”, “oxidoreductase activity”, “transferase activity”, “transcription regulator activity”, “integral component of membrane”, “bacterial-type flagellum”. None of the differentially-expressed proteins were found to be the same as previously reported biomarkers of non-culturable *L. pneumophila* [22]; however, there was overlap in many of the functional categories, such as “transmembrane transporter activity” and “integral component of membrane”.

Differences in protein abundances between the Cu-exposed and control conditions at t = 4 h were compared through LefSe analysis. Among all detected proteins, 110 and 86 were significantly enriched (*p* < 0.05, LDA > 2.0; Kruskal–Wallis tests) by the control or Cu-exposed condition, respectively. Cluster of Orthologous Groups (COG) categories were determined for those significantly enriched proteins (Figure 4, Appendix A). Cu dose increased the proportions of enriched proteins within COG categories of “intracellular trafficking and secretion”, “post-translational modification, protein turnover, chaperone functions”, “cell motility”, and “cell wall/membrane/envelop biogenesis”. Meanwhile, the Cu dose decreased the proportions of enriched proteins with COG categories of “energy production and conversion”, as well as primary metabolisms (Figure 4).

To shed light on mechanisms of inactivation imparted by Cu, proteins generated by Cu-exposed and control *L. pneumophila* were mapped against manually-curated GO ID databases to select those related to Cu response and oxidoreductive process. Similarly, proteins related to amoebae interactions and infectivity were selected for further analysis of *L. pneumophila* incubated with amoebae after Cu exposure (Appendix A). Most proteins included in the analysis did not significantly differ in abundance between the Cu-exposed and control *L. pneumophila*. In response to Cu exposure, two out of twelve proteins of interest indicated statistically significant differences, but one increased with copper exposure, and one decreased. Similar results were also observed in amoebae interaction and infectivity-related proteins (Appendix A). In oxidoreductive process-related proteins, one protein, Alkyl hydroperoxide reductase (AhpD) (Accession Number: A0A4Q5NC85, Appendix A), displayed significantly higher abundance in Cu-exposed *L. pneumophila*. Previous research indicated that AhpC (another subunit for alkyl hydroperoxide reductase) facilitated *Legionella* survival under oxidative stress [22]. However, there was no significant difference in abundance among the remaining eight proteins of interest.

### 3.4. Co-Incubation with Amoebae

Control *L. pneumophila* was able to replicate in the presence of *A. castellanii* at MOIs of 1:100 down to 1:0.01. At an MOI of 1:0.001, 50% of strain 130 b replicates produced increased levels of culturable *L. pneumophila*, but none of the outbreak-associated strain replicates achieved measurable re-growth. This was likely related to the lower initial density of culturable cells for the outbreak strain, which resulted in <1 CFU/mL for an MOI of 1:0.001. This established a minimum number of viable CFU inoculated into the amoebae co-incubation assay that resulted in growth for the outbreak-associated strain, but a similar minimum for strain 130 b could not be established in this study.

For the *L. pneumophila* outbreak strain, all Cu-dosed conditions yielded an initial level of culturable *L. pneumophila* < 1 CFU/mL. Because this is less than the threshold at which growth was also observed in the control, the findings are suggestive of resuscitation of Cu-induced VBNC cells by co-incubation. Although the *L. pneumophila* strain 130 b Cu-dosed conditions rendered culturable *L. pneumophila* levels (~2 CFU/mL) near 1 CFU/mL, it cannot be ruled out that a portion of culturable *L. pneumophila* was contributing to the observed culturability increase in the Cu-exposed conditions.

For strain 130 b, the EDTA washing procedure corresponded with observed growth in the Cu-exposed *L. pneumophila*, i.e., 43% of the time (3 out of 7 cases), with average culturable *L. pneumophila* increasing > two-log after 7-day co-culture. A longer co-culture time did not produce more *L. pneumophila* growth. No growth was observed in the EDTA complexing treatment. For the outbreak-associated strain, EDTA complexing resulted in 14% (1 out of 7) cases with observed *L. pneumophila* growth in the Cu-exposed condition, with only an average one-log increase in culturability. There was no growth in the EDTA washing conditions. The low observed growth of the outbreak strain under co-culture conditions could potentially be a result of (1) low MOI ratio of outbreak-associated strain co-culture compared with strain 130 b (1:10 vs. 1:100), which could reduce the infectivity/virulence of Cu exposed *L. pneumophila* and (2) long period of high level of Cu (5 mg/L for 24 h) exposure, which could drastically decrease the actual number of viable cells, given that VBNC status can be transient [33] (Figure 5).

## 4. Discussion

This study advances understanding of the effects of Cu on *L. pneumophila.* Although *L. pneumophila* lost culturability following Cu exposure, the suite of viability measures that we applied here was consistent with a proportion of the cells remaining viable. However, it was difficult to demonstrate consistently if putative VBNC cells defined based on the activity assays could be resuscitated by amoebae co-culture. Similar discrepancies were reported previously with monochloramine, UV disinfection, and heat treatment [3,4,6,16,19,20]. We conclude that activity-based assays thus provide an upper-bound estimate of the viability, whereas the amoebae co-culture method provides a lower-bound or more conservative estimate of VBNC cells. In other words, this study indicated that the viability tests treated as a proxy of infectivity might overestimate the actual levels of infective *L. pneumophila* after losing culturability. Importantly, this was an in vitro study, and it is likely that the variation in environmental conditions, amoebae species, and other factors could play an important role in shaping the response to copper.

We evaluated our findings with respect to those reported in prior studies of VBNC *Legionella* [14,34] to develop a conceptual model of the relationships among various measures of viability (Figure 6). Among them, co-culture with amoebae and mammalian macrophages not only demonstrates viability but also infectivity and, thus, is the most direct indicator of public health risk. Cell activity-related indicators, such as enzyme activity and ATP production, have been demonstrated to be consistent with cell viability [14] but could potentially overpredict the proportion of cells that are infective [4]. We also consider that the cell membrane integrity-based tests provide an upper-bound estimate of true infectivity as they provided the smallest log reduction in this study (Figure 2) and indicated a weak correlation between ATP levels and enzyme activity [14].

Including two distinct strains in this study highlighted that the mechanistic action of Cu varied among strains. Consistent with the outbreak-associated strain being previously demonstrated to be more resistant to copper [32], it was also found to be more difficult to grow in amoebae co-culture assays following Cu exposure. For example, although the proxy viability parameters indicated that there were >four-log higher viable cells than the culturable cells after Cu exposure, no growth was observed in amoebae co-culture in 13 out of 14 replicates. One contributing factor to this result could be that longer Cu-exposure times were required to reduce culturability in the Cu-exposed condition, resulting in greater lingering effects of copper.

Interestingly, under the same Cu exposure conditions, there were variable results among replicates of amoeba co-culture tests. For instance, *Legionella* Strain 130 b, presumed to be VBNC based on activity tests, only demonstrated growth in 43% of replicate amoeba co-culture tests (n = 7) after EDTA washing. This may be due, in part, to inherent variations in culturable/viable cells that are amplified during amoebae co-culture. Pinto and colleagues [33] indicated that VBNC cells could be in a transient state between fully healthy and dead cells. This may further contribute to the resuscitation result discrepancy because the transient state of the viable state in different test runs might dramatically affect the actual viable cells introduced for co-culture tests. In this study, the MOI ratios applied in the resuscitation of the outbreak-associated strain and strain 130 b were 1:10 (amoeba: *Legionella*) and 1:100, respectively. In other words, the outbreak-associated strain appeared to be more difficult to resuscitate by amoeba co-culture than strain 130 b. A likely contributing factor was that less culturable cells were introduced during the resuscitation experiments for the outbreak-associated strain experiments. Also, much longer exposure times of 24 h were required to reduce the culturability of the outbreak-associated strain, in addition to high levels of Cu, which could have reduced the chances that the non-culturable *Legionella* could maintain their infectivity. Additionally, the physiological response to Cu varies among different *Legionella* strains and might also play a role in the observed differences in their infectivity and resistance to *A. castellanii*. Therefore, amoebae co-culture methods would potentially benefit from optimization for specific strains of *Legionella*. Additional testing on means to ensure that copper antimicrobial activity is neutralized before co-culture would also be beneficial.

To our knowledge, this is the first study to apply whole-cell proteomics as a means to assess the mechanistic effects of Cu exposure on *Legionella*. Here we identified 32 proteins uniquely expressed in the Cu-exposed condition and eight additional proteins that were no longer expressed relative to the control. The unique proteins identified in the Cu-exposed *Legionella* condition include several related to the functions of “transport”, “transmembrane transporter activity”, and “integral component of membrane”, which were demonstrated to be biomarkers of non-culturable *L. pneumophila*. Notably, analysis of specific COG categories suggested that Cu-induced putative VBNC *L. pneumophila* had increased levels of proteins involved in the cell membrane, cell motility, and intracellular trafficking and secretion but decreased levels of proteins related to cellular metabolic processes. Lu et al. found that the expression of *L. pneumophila* genes involved in metabolism, transcription, translation, and DNA replication and repair was significantly affected by exposure to CuO nanoparticles, although the cells used in transcriptome analysis were not identified as putative VBNC state [35]. These observations support the hypothesis that Cu acts mechanistically to modulate cell membrane structure/function as well as core metabolic processes. The proteomics analysis using GO IDs indicated no significant changes of oxidative stress-related proteins between Cu-exposed and control *L. pneumophila,* other than one protein, Alkyl hydroperoxide reductase AhpD, which was enriched in the putative Cu-induced VBNC condition. AhpC was reported in a previous study to play a role in *Legionella* survival under oxidative stress [36]. Ameh and colleagues indicated increased levels of reactive-oxygen species in E. coli after exposure to Cu nanoparticles [37].

Additional research to understand how *L. pneumophila* responds to multiple, compounded environmental stresses could be of interest. For example, resistance to stresses, such as copper, and overall infectivity could conceivably increase with sequential exposure, infection of amoebae hosts, and resuscitation events [9]. However, one prior study evaluated copper resistance of *L. pneumophila* after intracellular growth and release from an amoebae host and found no difference in susceptibility between prior- and post-amoebae infection *Legionella* cells [38]. Nonetheless, Cirillo and colleagues [39] also claimed that the *Legionella* released from amoeba showed increased levels of virulence.

## Figures and Tables

**Figure 1 pathogens-13-00563-f001:**
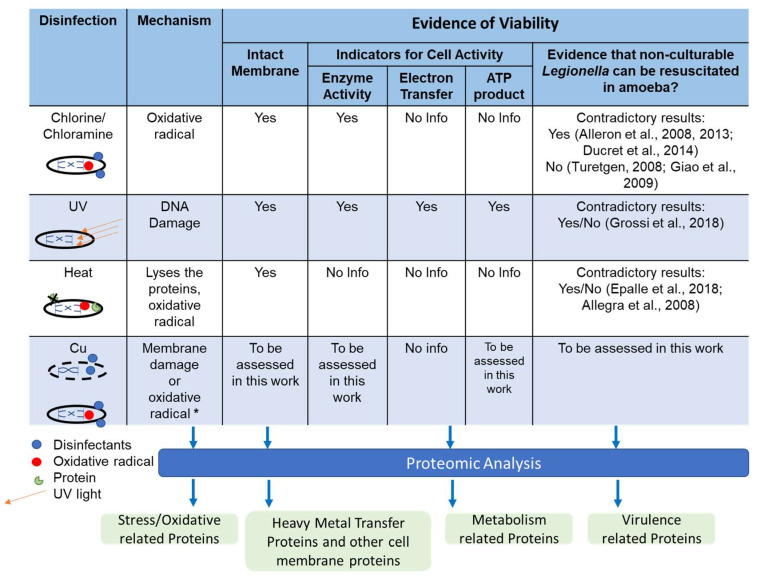
Research gaps regarding effects of copper on *Legionella*, including mechanism of lethal action and potential to induce a VBNC state. All listed disinfection methods have been demonstrated to be capable of fully eliminating culturability of *Legionella* [2,4,6,7,10,13,16,19]. * These are not all the potential mechanisms of Cu disinfection but represent the most common.

**Figure 2 pathogens-13-00563-f002:**
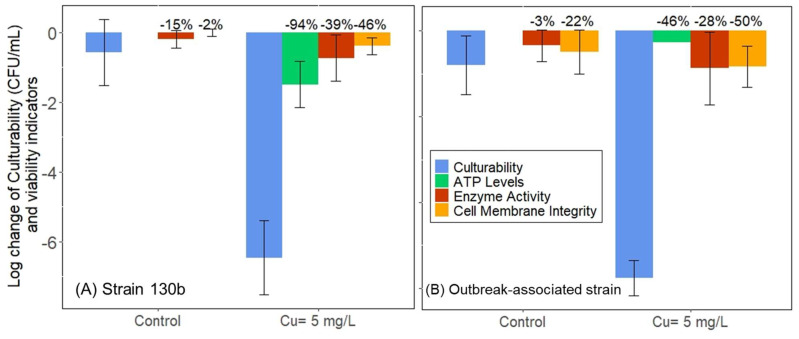
Log changes of *L. pneumophila* culturability (CFU/mL) and viability based on membrane integrity (%), enzyme activity (%), and ATP production (µM) for strain 130 b (**A**) and the outbreak-associated strain (**B**) with min. three biological replicates. Percent reduction in the measurement over the incubation period (4 h for Strain 130 b, 24 h for the outbreak-associated strain) is indicated for each measurement.

**Figure 3 pathogens-13-00563-f003:**
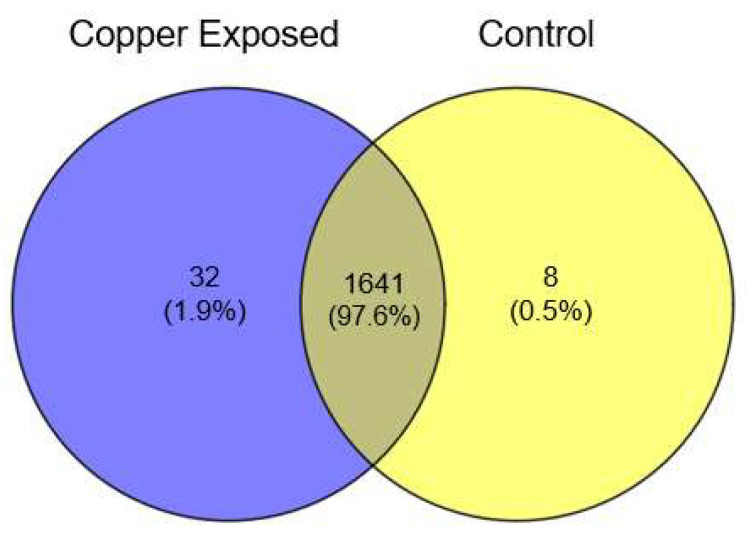
Proteins detected in copper-exposed and control *L. pneumophila* strain 130 b. Proteins/peptides were detected by mass spectrometry and identified using *L. pneumophila* proteome database. Numbers represent unique proteins identified in at least two of the three replicates.

**Figure 4 pathogens-13-00563-f004:**
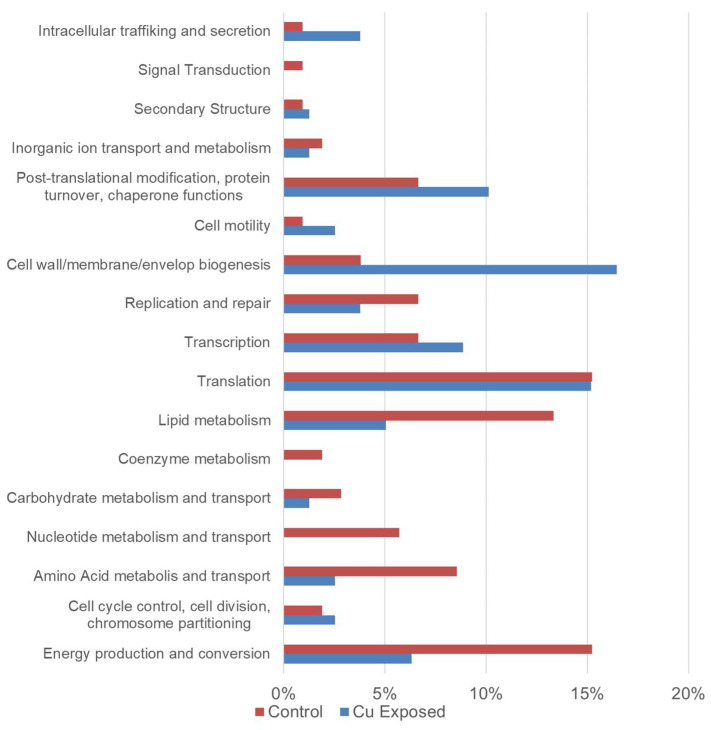
Cluster of orthologous groups (COG) categories percentile among proteins that were significantly higher in abundance in the copper = 5 mg/L (Blue) and control (maroon) *L. pneumophila* Strain 130 b after 4 h incubation. Percentages were calculated by dividing the enriched proteins categorized into the indicated COG category by the total enriched proteins in control or Cu-exposed conditions.

**Figure 5 pathogens-13-00563-f005:**
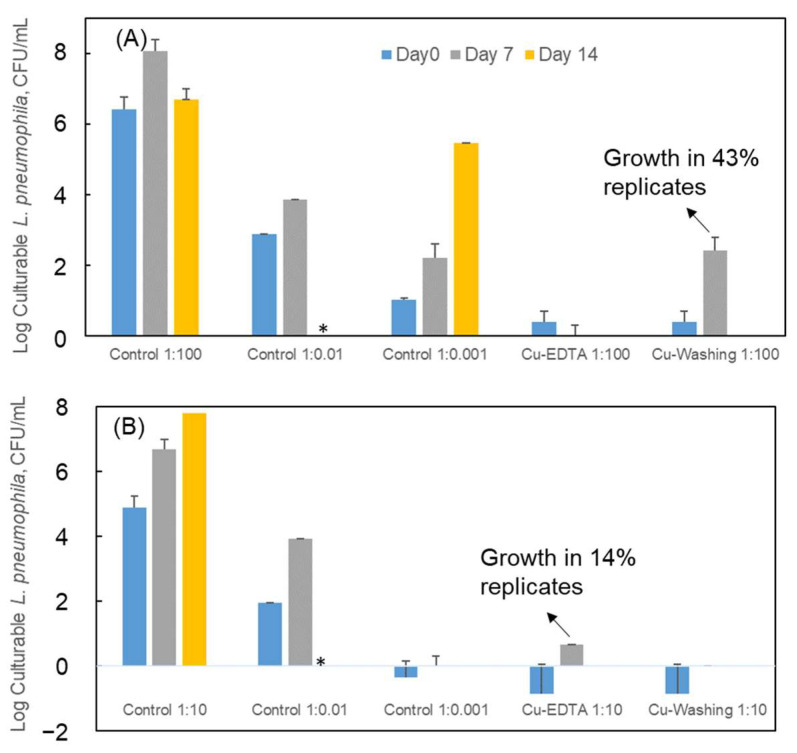
Culturability of (**A**) *L. pneumophila* strain 130 b and (**B**) outbreak-associated strain after co-culturing with *A. castellanii* for 0, 7 and 14 days. Control (no copper) was co-cultured at multiplicities of infection (MOIs) of 1:100 to 1:0.001 (amoebae: *L. pneumophila*) and copper-exposed at MOIs of 1:100 to 1:10. Effects of Cu remaining in solution interfering with assay were mitigated by directly mixing with EDTA at 1:5 (Cu: EDTA) molar ratio (Cu-EDTA) or mixing with the same EDTA solution, incubating for 30 min, and removing > 90% copper from the cells through centrifugation and resuspension (Cu- washing). *—No samples collected. Note, four biological replicates with three technical replicates were included in each round of experiment.

**Figure 6 pathogens-13-00563-f006:**
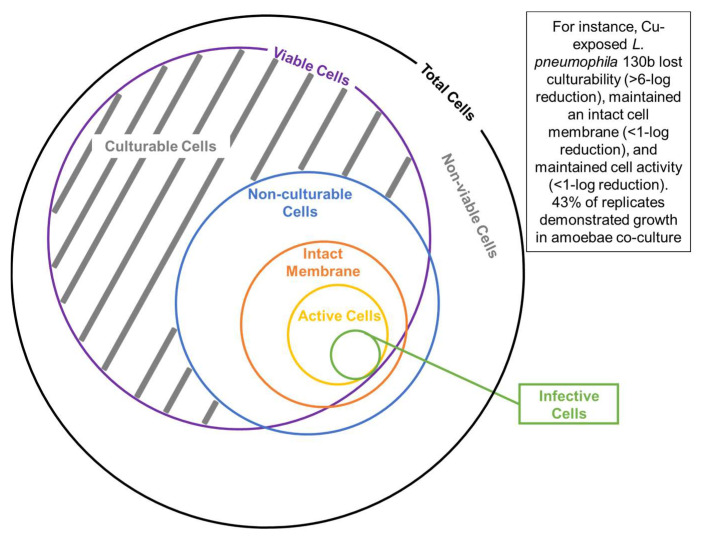
Conceptual relationship among total cells, viable cells, culturable cells, non-culturable cells, cells with intact membrane, cells maintaining cell activity, and cells capable of being resuscitated by amoeba co-culture (infectivity). The sizes of the subcomponents do not imply quantitative relationships.

## Data Availability

Data supporting reported results can be presented upon reasonable request.

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
