# Peer review of "Effects of Copper on Legionella pneumophila Revealed via Viability Assays and Proteomics"

_pathogens, 2024, doi:10.3390/pathogens13070563_

Round 1
Reviewer 1 Report
Comments and Suggestions for Authors
The manuscript describes the response of two strains of Legionella pneumophila to copper. First, phenotypic analysis was performed via CFU counts and viability assay, showing that copper exposure seems to induce L. pneumophila into a VBNC state. Second the proteome of one strain, with and without copper, is defined. Finally, the authors attempt to resuscitate VBNC cells in amoeba.
Overall, the paper provides interesting data about the effect of copper on Legionella. It is well and clearly structured. Nevertheless, the paper consists of 3 main parts that are covered only superficially, as detailed below, which fall short of the authors’ stated objectives. Possible improvements are suggested below and the discussion should be improved accordingly.
General comments:
1. The first line of discussion state: “This study advances mechanistic effect of Cu on Legionella pneumophila”; however, in this reviewer opinion, the data presented is insufficient to make this claim. No putative mechanism can be derived from the study, except a minor induction of aphD, which involvement was not confirmed experimentally. It is recommended to remove statement pertaining to mechanism of copper effect in Legionella.
2. Figure present the survival and viability of two different strain of L. pneumophila. The authors used different experimental parameters for each strain (4h/24h incubation and a different density of cells) preventing direct comparison between the strain. It is suggested to show data for both time points for both strains. In addition, cell density is well known to affect the response of microbes to stressors. Therefore, one should use the same density when comparing strains.
3. The authors attempt resuscitation if VBNC in A. castellanii. It is customary in the field to always include a control strain defective for intracellular multiplication when doing infection, since L. pneumophila is sometime able to grow extracellularly in the infection medium.
4. The authors should be more explicit about the number of replicates included in each figure and their nature (biological or technical replicate).
5. The proteomic experiment is very interesting. The authors identified 32 proteins unique to the copper condition, and 8 unique to the control condition. More details should be provided about those 40 proteins, including gen name and putative pathway they are involved in. the author should supply this information in a table in the manuscript. In addition, the authors should include lpg#### (or equivalent gene number in 130b) and gene name in the supplementary table reporting proteomic data. Line 348, accession number for aphD should be provided).
6. COG analysis should be improved as several categories do not make sense for a bacteria, including “cell trafficking” and “mitosis”
7. Introduction should be improved to include relevant knowledge and citations about copper response in Legionella and other bacterial species. The work of Bédard 2021 and Lu 2013 should be cited, and the later might be useful to interpret the proteomic data.
Specific comments:
Line 81: remove “other” as plastic pipe are not the same as copper pipe.
Figure 1: change “to access” to “to be assessed”
Figure 1: several other mechanisms of copper action have been described which could be included in the figure.
L152: missing a bracket.
L177: “cell” not “cells”
L247: “Interesting” not “interested”
Figure 4: “translation” not tranlsation”, “cell trafficking” not “cell trafficing”
Line 486: an example of oxidative stress in bacteria would be more appropriate here.
Comments on the Quality of English LanguageA few typos are noted above but this is not an exhaustive list and it suggested that the text be reviewed extensively.
Author Response
Response to Review of Song et al. “Effects of Copper on Legionella pneumophila Revealed via Viability Assays and Proteomics”. Please note the line numbers refer to the tracked changes version of the manuscript.
Reviewer 1.
Comment 1-0 The manuscript describes the response of two strains of Legionella pneumophila to copper. First, phenotypic analysis was performed via CFU counts and viability assay, showing that copper exposure seems to induce L. pneumophila into a VBNC state. Second the proteome of one strain, with and without copper, is defined. Finally, the authors attempt to resuscitate VBNC cells in amoeba.
Overall, the paper provides interesting data about the effect of copper on Legionella. It is well and clearly structured. Nevertheless, the paper consists of 3 main parts that are covered only superficially, as detailed below, which fall short of the authors’ stated objectives. Possible improvements are suggested below and the discussion should be improved accordingly.
We thank the reviewer for the comment. We have addressed the following comments and improved the manuscript to refine the stated study objective and extend the discussion accordingly. Please see the detailed response to each comment below.
Comment 1-1 The first line of discussion state: “This study advances mechanistic effect of Cu on Legionella pneumophila”; however, in this reviewer opinion, the data presented is insufficient to make this claim. No putative mechanism can be derived from the study, except a minor induction of aphD, which involvement was not confirmed experimentally. It is recommended to remove statement pertaining to mechanism of copper effect in Legionella.
We acknowledge the reviewer’s comment and have updated the first sentence in discussion to be “This study advances understanding of the effects of Cu on L. pneumophila” in line 435.
Comment 1-2 Figure present the survival and viability of two different strain of L. pneumophila. The authors used different experimental parameters for each strain (4h/24h incubation and a different density of cells) preventing direct comparison between the strain. It is suggested to show data for both time points for both strains. In addition, cell density is well known to affect the response of microbes to stressors. Therefore, one should use the same density when comparing strains.
We agree with the reviewer that the initial cell density and exposure time will affect the response of microorganisms to copper disinfection. Based on our previous study (Song et al. 2023), the outbreak-associated strain has a much higher level of copper resistance than strain 130b. Since part of the study goal is to induce viable but non-culturable L. pneumophila through copper disinfection, we would like to minimize the culturability after copper disinfection for both strains. For strain 130b, based on previous study (Song et al., 2023) and multiple test runs, we found the combination of ~3x107 CFU/mL initial cell density, 4-hr incubation time and 5 mg/L Cu at pH=6.5 would typically render <1 CFU/mL culturable. While the outbreak-associated strain would require longer copper exposure time (24hr) and less initial cell density (~3x106 CFU/mL) at the same concentration of copper dose. By adjusting the initial cell density and exposure time for the outbreak associated strain, we were able to maintain the same level of copper dose and reach similar culturability levels following exposure. To further clarify this, we have updated the sentence in lines 153-155 as “… viable levels of L. pneumophila at t=4, 24 for 130b and outbreak strain (sampling intervals and initial cell densities were determined based on previous test runs to achieve <1 CFU/mL culturability).”
Comment 1-3 The authors attempt resuscitation if VBNC in A. castellanii. It is customary in the field to always include a control strain defective for intracellular multiplication when doing infection, since L. pneumophila is sometime able to grow extracellularly in the infection medium.
We appreciate the reviewer’s comment. We included positive (healthy L. pneumophila) and negative (only A. castellanii) controls for the putative VBNC L. pneumophila resuscitation experiments. We have added this information in the coculture methods section in lines 281-282. We did not include a control strain that is defective for intracellular multiplication in our coculture experiments. We understand that the reviewer is concerned that putative VBNC L. pneumophila could grow extracellularly in the PYG medium. To our knowledge, we have not found any evidence that L. pneumophila could grow in PYG medium (Mou & Leung, 2017).
Comment 1-4 The authors should be more explicit about the number of replicates included in each figure and their nature (biological or technical replicate).
We have updated the captions of Figure 2 and 5 to clearly indicate the replicates and their nature. Please see lines 327 and 421-422.
Comment 1-5 The proteomic experiment is very interesting. The authors identified 32 proteins unique to the copper condition, and 8 unique to the control condition. More details should be provided about those 40 proteins, including gen name and putative pathway they are involved in. the author should supply this information in a table in the manuscript. In addition, the authors should include lpg#### (or equivalent gene number in 130b) and gene name in the supplementary table reporting proteomic data. Line 348, accession number for ahpD should be provided).
We thank the reviewer for the comments. We now provide further detailed information in supplemental information SI-3 regarding proteins uniquely found in copper exposed and control conditions with accession number, description and putative pathway they are involved in. Regarding the accession number for ahpD, it was included in the supplemental information table SI-5. We have also highlighted the information in the manuscript in lines 361-362 as “In oxidoreductive process related proteins, one protein, Alkyl hydroperoxide reductase (AhpD) (Accession Number: A0A4Q5NC85, Table SI-5)…”
Comment 1-6 COG analysis should be improved as several categories do not make sense for a bacteria, including “cell trafficking” and “mitosis”.
We thank the reviewer for the comment. COG category U includes intracellular trafficking, secretion, and vesicular transport. There were three and one proteins enriched in the copper exposed and control conditions, respectively, that were categorized as COG category U. All four proteins (A0A2S6F779, A0A378KD96, A0A128ZHV9, and A0A378KFV9) are involved in the secretion process rather than only intracellular trafficking. As a matter of fact, bacterial pathogens, including L. pneumophila, could alter the host plasma membrane, and hijack various vesicle trafficking pathways to ensure survival through intracellular trafficking and secretion.
|
Access Number |
Enriched in Cu exposed or control conditions |
Description |
|
A0A378KFV9 |
Control |
Part of the Sec protein translocase complex. Interacts with the SecYEG preprotein conducting channel. Has a central role in coupling the hydrolysis of ATP to the transfer of proteins into and across the cell membrane, serving both as a receptor for the preprotein-SecB complex and as an ATP-driven molecular motor driving the stepwise translocation of polypeptide chains across the membrane |
|
A0A2S6F779 |
Cu exposed |
involved in the tonB-independent uptake of proteins |
|
A0A378KD96 |
Cu exposed |
Outer membrane efflux protein |
|
A0A128ZHV9 |
Cu exposed |
Involved in targeting and insertion of nascent membrane proteins into the cytoplasmic membrane. Binds to the hydrophobic signal sequence of the ribosome-nascent chain (RNC) as it emerges from the ribosomes. The SRP-RNC complex is then targeted to the cytoplasmic membrane where it interacts with the SRP receptor FtsY. Interaction with FtsY leads to the transfer of the RNC complex to the Sec translocase for insertion into the membrane, the hydrolysis of GTP by both Ffh and FtsY, and the dissociation of the SRP-FtsY complex into the individual components |
Regarding the comment on mitosis, it is agreed that it does not make sense for bacteria to be categorized under mitosis. Therefore, we updated the category name to “Cell cycle control, cell division, chromosome partitioning”. Please see updated Figure 4. In addition, we have confirmed that all those proteins that were identified as COG category D – Cell cycle control, cell division, chromosome partitioning actually exist in bacteria region as shown in the table below.
|
Access Number |
Enriched in Cu exposed or control conditions |
Description |
|
A0A4Q5NDR8 |
Control |
nuclear chromosome segregation |
|
A0A130QU18 |
Control |
Cell division protein that is involved in the assembly of the Z ring. May serve as a membrane anchor for the Z ring |
|
A0A4Q5NEF4 |
Cu exposed |
Mediates coordination of peptidoglycan synthesis and outer membrane constriction during cell division |
|
A0A378KFM9 |
Cu exposed |
Involved in protein export. Acts as a chaperone by maintaining the newly synthesized protein in an open conformation. Functions as a peptidyl-prolyl cis-trans isomerase |
Comment 1-7 Introduction should be improved to include relevant knowledge and citations about copper response in Legionella and other bacterial species. The work of Bédard 2021 and Lu 2013 should be cited, and the later might be useful to interpret the proteomic data.
We appreciate the reviewer’s comment. We have now included the work of Bédard 2021 et al. titled as “Local Adaptation of Legionella pneumophila within a hospital hot water system increases tolerance to copper” as well as the other article “Legionella pneumophila Transcriptional Response following Exposure to CuO Nanoparticles” by Lu et al. in 2013 with associated introduction and discussion updated in lines 73-74 and 496-499.
Comment 1-9 Line 81: remove “other” as plastic pipe are not the same as copper pipe.
The word “other” is now removed.
Comment 1-10 Figure 1: change “to access” to “to be assessed”
The wording in Figure 1 is now updated as “to be assessed”. Please see revised Figure 1 in line 110.
Comment 1-11 Figure 1: several other mechanisms of copper action have been described which could be included in the figure.
We assume the reviewer is referring to some other mechanisms of copper disinfection. Figure 1 is designed to serve specifically for this study with highlights of research gaps in copper disinfection towards L. pneumophila. We now updated the caption in Figure 1 to honor other possible mechanisms of copper disinfection as stated in lines 113-114, “* These are not all the potential mechanisms of Cu disinfection but represent the most common.”
Comment 1-12 L152: missing a bracket.
We thank the reviewer for this comment. However, we have not identified the missing bracket location on pages 3-5. We can address this minor issue in the final editing process if it’s missed.
Comment 1-13 L177: “cell” not “cells”
The wording has been updated.
Comment 1-14 L247: “Interesting” not “interested”
The wording has been updated to be “Protein abundances within the categories of interest were compared between Cu exposed nonculturable and Cu free culturable L. pneumophila conditions.” in lines 260-261.
Comment 1-15 Figure 4: “translation” not tranlsation”, “cell trafficking” not “cell trafficing”
The wording in Figure 4 is now updated as “translation” and “Intracellular trafficking and secretion”. Please see revised Figure 4 in line 375.
Comment 1-16 Line 486: an example of oxidative stress in bacteria would be more appropriate here.
We thank the reviewer for the comment. An example of copper induced oxidative stress in E. coli is now referenced as stated in lines 505-506: “Ameh and colleagues indicated increased levels of reactive-oxygen-species in E. coli after exposure to Cu nanoparticles”.
Comment 1-17 Comments on the Quality of English Language. A few typos are noted above but this is not an exhaustive list and it suggested that the text be reviewed extensively.
We have thoroughly checked the language again to improve the clarity for this manuscript. Please see changes in responses from comment 1-10 to comment 1-15.
References
Mou, Q.; Leung, PHM. Differential expression of virulence genes in Legionella pneumophila growing in Acanthamoeba and human monocytes. Virulence 2018, Jan 1;9(1):185-196. doi: 10.1080/21505594.2017.1373925.
Song, Y.; Pruden, A.; Rhoads, W. J.; Edwards, M. A. Pilot-scale assessment reveals effects of anode type and orthophosphate in governing antimicrobial capacity of copper for Legionella pneumophila control. Water Research 2023, 242, 120178. https://doi.org/10.1016/j.watres.2023.120178
Cullom, A.C.; Martin, R.L.; Song, Y.; Williams, K.; Williams, A.; Pruden, A.; Edwards, M.A. Critical Review: Propensity of Premise Plumbing Pipe Materials to Enhance or Diminish Growth of Legionella and Other Opportunistic Pathogens. Pathogens 2020, 9, 957. https://doi.org/10.3390/pathogens9110957
Reviewer 2 Report
Comments and Suggestions for Authors
Dear author's, after reading your paper I have few question about:
1) what is the novelty of the work? Could You describe it wider on base lines 83-90?
2) please add comment to describes methods - are they standard or modified by Author's?
Author Response
Response to Review of Song et al. “Effects of Copper on Legionella pneumophila Revealed via Viability Assays and Proteomics”. Please note the line numbers refer to the tracked changes version of the manuscript.
Reviewer 2.
Comment 2-1 what is the novelty of the work? Could You describe it wider on base lines 83-90?
The novelty of this work is multi-faceted: 1) to our knowledge this is the first study to specifically test whether copper disinfection could induce L. pneumophila as VBNC state, which is highlighted in lines 93-95; 2) Multiple metabolic assays were implemented to assess the viability of L. pneumophila in the copper disinfection analysis, which is highlighted in lines 47-58; and 3) this study pioneers the examination of effects of copper on L. pneumophila using proteomics, which is highlighted in lines 75-84.
Comment 2-2 please add comment to describes methods - are they standard or modified by Author's?
We have provided references to the methods used in this study as described in the Methods section. All study approaches followed the referenced methods strictly and any modifications are clearly described in the manuscript as well. For specific experimental methods that are newly proposed in this study such as “cell washing to remove Cu using EDTA solution”, the methods were described in detail to make sure they are reproducible by other researchers.